

# An efficient hybrid downscaling framework to estimate high-resolution river hydrodynamics

Zeli Tan[1], Donghui Xu[1], Sourav Taraphdar[1], Jiangqin Ma[2], Gautam Bisht[1], L. Ruby Leung[1]

[1]Pacific Northwest National Laboratory, Richland, 99352, USA
[2]College of Engineering, Georgia Institute of Technology, Atlanta, 30332, USA

*Correspondence to*: Zeli Tan (zeli.tan@pnnl.gov)

**Abstract.** Flow depth and velocity are the most important hydrodynamic variables that govern various river functions, including water resources, navigation, sediment transport, and biogeochemical cycling. Existing high-resolution flow depth simulations rely on either computationally expensive river hydrodynamic models (RHMs) or data-driven models with
formidable training costs, whereas data-driven modeling of flow velocity has rarely been explored. Here, using the hybrid Low-fidelity, Spatial analysis, and Gaussian Process learning (LSG) model, we developed a downscaling approach to accurately construct high-resolution flow depth and velocity from a two-dimensional (2-D) RHM simulation at coarse resolution. The LSG models were trained and tested in an urban watershed in Houston using two different hurricane-driven flood events. The results showed that through downscaling, the simulation errors were reduced to less than one-fourth and one-
third of the errors of the low-resolution 2-D RHM for flow depth and velocity, respectively. Our analysis further revealed that the dominant uncertainty sources of the downscaled hydrodynamics are different, with flow velocity dominated by the dimensionality reduction error, which we reduced by using a regionalized training procedure. The downscaling approach achieves an 84-fold acceleration in computational time compared to the high-resolution 2-D RHM, making high-fidelity ensemble flood modeling feasible. More importantly, the developed method provides an opportunity to couple large-scale
hydrodynamical processes with local physical, chemical, and biological processes in river models.

## 1 Introduction

Rivers play a crucial role in water resources, navigation, sediment transport, and biogeochemical cycling (Syvitski et al., 2005; Oki & Kanae, 2006; Allen & Pavelsky, 2018; Ibáñez & Peñuelas, 2019; Mao et al., 2019; Regnier et al., 2022; Feng et al., 2023a; Rocher-Ros et al., 2023). To sustain these vital services, river flow depth and velocity must remain within normal
ranges. Extreme flow depths can result in extensive fluvial flooding (Bates, 2022), whereas prolonged low flow depths jeopardize the availability of drinking and irrigation water in many regions worldwide (Gadgil, 1998; Haddeland et al., 2006). Together, flow depth and velocity are key drivers of navigation capability, sediment transport, and biogeochemical processes in rivers (Zhang et al., 2014; Raymond et al., 2016; Li et al., 2022; Sukhodolov et al., 2023). Consequently, extreme variations in flow depth and velocity can lead to waterway blockage, channel aggradation or degradation, water quality deterioration, and





habitat loss. River flow depth and velocity regimes are dynamic and influenced by climate change and human activities, leading many rivers to experience extreme flow conditions (Mishra & Shah, 2018). These conditions exacerbate flooding (Freer et al., 2011), degrade aquatic ecosystems (Carpenter et al., 2011; Battin et al., 2023), and diminish water supplies (Oki & Kanae, 2006). Hence, accurate prediction of river flow depth and velocity in the context of a changing climate is essential for ensuring the well-being of human society (IPCC, 2021).

Flow depth and velocity are commonly simulated using river hydrodynamic models (RHMs). Widely used RHMs are often based on one-dimensional (1-D) or two-dimensional (2-D) Saint-Venant equations, disregarding vertical variations due to the significant difference between the horizontal and vertical length scales of rivers (Li et al., 2013; Teng et al., 2017; Bates, 2022; Huang et al., 2022). Considering the low computational cost and high numerical stability, Earth system models (ESMs) usually employ 1-D RHMs as the river component for large-scale and/or ensemble hydrological simulations (Li et al., 2013;

Feng et al., 2024). However, they are unsuitable for high-fidelity flood simulations. This is because 1-D RHMs are solved on upscaled river networks rather than actual river reaches (Wu et al., 2011; Liao et al., 2022) and rely on uncertain parameterizations, such as the bathtub method for estimating floodplain inundation (Luo et al., 2017; Xu et al., 2022). Additionally, by oversimplifying and/or neglecting momentum transport in river channels and floodplains (Luo et al., 2017; Feng et al., 2022), 1-D RHMs lack the capability to simulate fine-scale river hydrodynamics required for geomorphological

and biogeochemical modeling (Hostache et al., 2014; Shabani et al., 2021). Conversely, 2-D RHMs can solve full river dynamics. When running on high-resolution meshes, they can accurately capture river flow depth and velocity (Razavi et al., 2012). Therefore, high-resolution 2-D RHMs are often referred to as high-fidelity (HF) models, whereas both 1-D RHMs and low-resolution 2-D RHMs are referred to as low-fidelity (LF) models. However, the significant computational cost of HF RHMs (Teng et al., 2017; Wu et al., 2020; Ivanov et al., 2021) makes them not viable for real-time modeling and flood risk

assessments through ensemble modeling, which requires hundreds or thousands of model realizations (Wu et al., 2020).

    To achieve accurate and affordable simulations of river hydrodynamics, several alternative approaches have been developed (Razavi et al., 2012). One prominent approach is the use of data-driven models to emulate the behaviors of HF RHMs (Ivanov et al., 2021; Tran et al., 2023). With the rapid advancement of machine learning (ML) techniques, ML-based emulators have been increasingly employed in hydrological sciences, including applications such as modeling runoff (Gao et

al., 2020), evapotranspiration (Hu et al., 2021), inundation (Xie et al., 2021), lake-river interactions (Liang et al., 2018; Huang et al., 2022), reservoir operations (Zhang et al., 2018; Yang et al., 2019), streamflow (Ha et al., 2021; Sikorska-Senoner & Quilty, 2021), groundwater (He et al., 2020; Wunsch et al., 2022), and water quality (Chen et al., 2020; Saha et al., 2023). These studies have demonstrated that, once trained under extensive conditions, the computationally efficient ML models can mimic numerical models. However, general ML-based emulators often lack the enforcement of physical laws, such as the

conservation of mass and momentum (Konapala et al., 2020; Karniadakis et al., 2021), resulting in poor transferability to out-of-sample conditions in nonstationary systems (Young et al., 2017; Konapala et al., 2020), such as streamflow in a changing climate. To address this limitation, variants like physics-informed neural networks have been developed, embedding physical





laws (e.g., Saint-Venant equations) into their cost functions to constrain ML solutions. However, the incorporation of physical laws tends to reduce the training efficiency of ML models (Feng et al., 2023b).

The second approach is to downscale the low-resolution RHM simulation onto a finely discretized grid (Wilby & Dawson, 2013; Feng et al., 2023b). For instance, Bermúdez et al. (2020) created high-resolution inundation maps by simply interpolating flow depth computed from a LF RHM onto a high-resolution digital elevation model (DEM). Recently, more advanced downscaling methods have been developed using various ML techniques to reproduce the detailed spatial and temporal features of high-resolution river hydrodynamics (Carreau & Guinot, 2021). Notably, Fraehr et al. (2022) developed

a novel downscaling method based on the hybrid Low-fidelity, Spatial analysis, and Gaussian Process learning (LSG) model. This method demonstrated promising accuracy in simulating the dynamic behavior of flood inundation, such as the rising and recession components and hysteresis, at the computational cost of a low-resolution 2-D RHM (Fhraehr et al., 2022). Later, Fraehr et al. (2023a) extended the approach for fast and accurate simulations of not only high-resolution flood extent but also high-resolution flow depth. Additionally, the LSG-based downscaling model can support both structured and unstructured

grids, a significant advantage as modern 2-D RHMs increasingly adopt unstructured grids for fine-scale modeling (Begnudelli & Sanders, 2006; Kim et al., 2012). However, similar to Fraehr et al. (2022 & 2023a), existing research on hydrodynamic model downscaling has focused entirely on flood extent and magnitude, while ignoring flow velocity, a critical factor for human safety risks in flood events (Russo et al., 2013). Moreover, as discussed earlier, without accurate simulations of flow velocity, our understanding of how river functions will respond to environmental stresses will remain elusive.

In this study, we develop an LSG-based downscaling approach to achieve accurate simulations of high-resolution river flow depth and velocity at the computational cost of a low-resolution 2-D RHM. The effectiveness and transferability of our method are tested in an urbanized watershed in the Houston area using data from two extreme hurricane events. Furthermore, based on this downscaling method, we propose a new paradigm to couple large-scale hydrodynamical processes with local detailed physical, chemical, and biological processes in river models. The remainder of this paper is organized as follows:

Section 2 describes the downscaling method and the configurations of the high-resolution and low-resolution 2-D RHMs for the study events; Section 3 highlights the main results; Section 4 discusses the implications of the results, outlines the limitations of our approach, and introduces the new paradigm; and Section 5 concludes the paper.

## 2 Materials and Methods

### 2.1 LSG model

For the LSG model, the underlying principle is that the dynamics of flow depth and velocity can be approximated by a limited number of temporal and spatial modes due to their strong spatial pattern controlled by topography. The LSG model consists of a LF RHM, key spatial modes extracted from a HF RHM, and a Sparse Gaussian Process (GP) emulator model. It uses the LF RHM as a transfer function to capture the dynamics and spatial correlation of river flow. The key temporal features of the LF RHM outputs are extracted through an Empirical Orthogonal Function (EOF) analysis based on the extracted spatial



features from the HF RHM, thereby allowing the use of a Sparse GP model to convert the LF data to HF data via conversion of the extracted temporal features. The LSG model can reconstruct high-fidelity river hydrodynamics for two reasons. First, the accurate spatial correlations of river hydrodynamics are preserved due to the use of the key spatial modes from the HF model, which are assumed to not vary by events. Second, the Sparse GP model is efficient and effective in reconstructing the dynamics of HF data. In this study, we used the same 2-D shallow water equations to construct the LF and HF RHMs (described

later). The only difference between them is the spatial resolution, with the coarse mesh adopted by the LF model reducing the simulation accuracy. We followed the procedure described by Fraehr et al. (2022, 2023a) to train and apply LSG models for river flow depth and velocity downscaling (Fig. 1), with any deviations from the general procedure specifically highlighted.

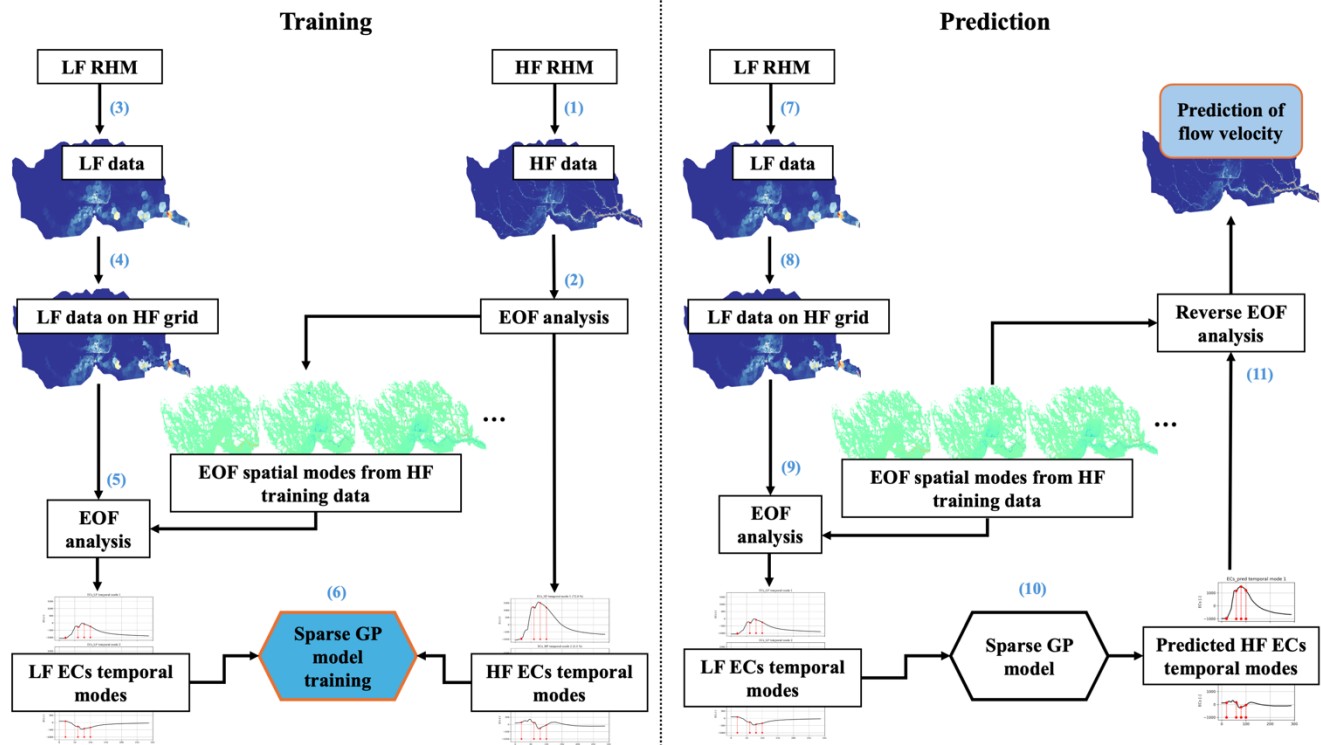

Figure 1: Workflow of training the LSG model and using the trained model to predict high-resolution river hydrodynamics.

For training, we first run the HF RHM over the study domain for a training flood event (Step 1) and derive the spatial EOF modes and the temporal expansion coefficients (ECs) modes of this HF simulation through the EOF analysis (Step 2) as defined in Eq. (1).

$$D_{HF} = U_{HF} \cdot C_{HF} \approx \sum_{k=1}^{K} U_{HF}(k,:) \cdot C_{HF}(:,k), \qquad (1)$$

where $D_{HF}$ is a $T \times N$ matrix containing simulated HF flow depth or velocity ($T$ is the number of timesteps in the training data

and $N$ is the number of wet cells) that have been detrended (Fraehr et al., 2023a), $U_{HF}$ is a $T \times N$ matrix each row of which is an EOF spatial map, $C_{HF}$ is a $T \times T$ matrix each column of which corresponds to an EC temporal function, and $K$ is the number of significant modes determined by both North's test (North et al., 1982) and Kaiser's Rule (Kaiser, 1960).





For the training phase, we also run the LF RHM for the training event (Step 3) and interpolate the simulated flow depth and velocity from the coarse mesh used by the LF model to the fine mesh used by the HF model (Step 4). Notably, we improved the nearest neighbor interpolation method adopted by Fraehr et al. (2023a) by accounting for mass conservation. While our improved method still assumes a homogeneous water level within a coarse grid cell, it ensures that the sum of interpolated water volume in fine grid cells equals the water volume in the coarse grid cell. Additionally, we ensure that the interpolation of flow velocity only occurs at wet grid cells where the water depth is greater than 3 cm (Fhraehr et al., 2023a). Another difference is that we do not apply area-based weights to $D_{HF}$ before the EOF analysis, as Fraehr et al. (2023a) recommended. This is because we are more interested in the flow depth and velocity of river channels and nearby floodplains that are represented by smaller grid cells in our fine mesh (Fig. 2).

Next, we perform the EOF analysis on the interpolated LF flow depth and velocity to derive the temporal EC modes of the LF simulation (Step 5). Using the extracted high-resolution EOF spatial modes from Step 2, the extracted temporal ECs are defined in Eq. (2).

$$C_{LF} = D_{LF} \cdot U'_{HF}, \tag{2}$$

where $C_{LF}$ is a $T \times T$ matrix containing the LF ECs, $D_{LF}$ is a $T \times N$ matrix corresponding to the interpolated LF flow depth or velocity simulations, and $U'_{HF}$ is the transpose of $U_{HF}$. In the final step of training, we use the derived LF and HF temporal ECs to train a Sparse GP model (Rasmussen & Williams, 2006) that can predict the HF ECs from the LF ECs (Step 6). For flow depth and velocity, the training of the Sparse GP models is performed independently.

For prediction, only the low-cost LF RHM simulations are needed (Fig. 1). While Steps 7 to 9 essentially replicate Steps 3 to 5, the difference is that Steps 7 to 9 are applied to a new LF simulation that is run for an unseen flood event. After the new LF ECs are retrieved following the EOF analysis in Eq. (1) using the spatial EOF modes derived in Step 2, they are fed into the trained Sparse GP model to predict the new HF ECs (Step 10). Finally, the predicted HF ECs are combined with the EOF spatial modes from Step 2 to reconstruct the HF flow depth and velocity simulations based on the reverse EOF analysis (Step 11) as defined in Eq. (3).

$$\widehat{D_{LSG}} = \sum_{k=1}^{K} U_{HF}(k,:) \cdot \widehat{C_{LSG}}(:,k), \tag{3}$$

where $\widehat{D_{LSG}}$ is the predicted high-resolution flow depth or velocity, and $\widehat{C_{LSG}}$ is the predicted HF temporal ECs. More details of the workflow can be found in Fraehr et al. (2022 & 2023a).

The downscaling error consists of two major components: the error from dimensionality reduction and the error from the LSG model. According to Eq. (1), the error from dimensionality reduction $ER_{DR}$ can be defined as: $ER_{DR} = D_{HF} - \sum_{k=1}^{K} U_{HF}(k,:) \cdot C_{HF}(:,k)$. According to Eq. (3), the error from the LSG model $ER_{LSG}$ can be defined as: $ER_{LSG} = \sum_{k=1}^{K} U_{HF}(k,:) \cdot C_{HF}(:,k) - \sum_{k=1}^{K} U_{HF}(k,:) \cdot \widehat{C_{LSG}}(:,k)$.



## 2.2 Study site and flood events

We used the Hurricane Harvey flood event (hereafter referred to as Harvey) in the Houston area as a case study. On
August 26, 2017, Harvey made landfall along the mid-Texas coast as a Category 4 hurricane. As one of the worst hurricanes
to hit the United States in recent history, Harvey brought record-breaking rainfall across the Houston metropolitan area (Van
Oldenborgh et al., 2017), causing more than 80 fatalities and over $150 billion in economic losses, mostly due to extraordinary
flooding (Emanuel, 2017; Balaguru et al., 2018). Specifically, we selected the Buffalo Bayou at Turning Basin as the study
domain (Fig. 2), where the selected RHM was recently validated at different resolutions (Xu et al., in review).

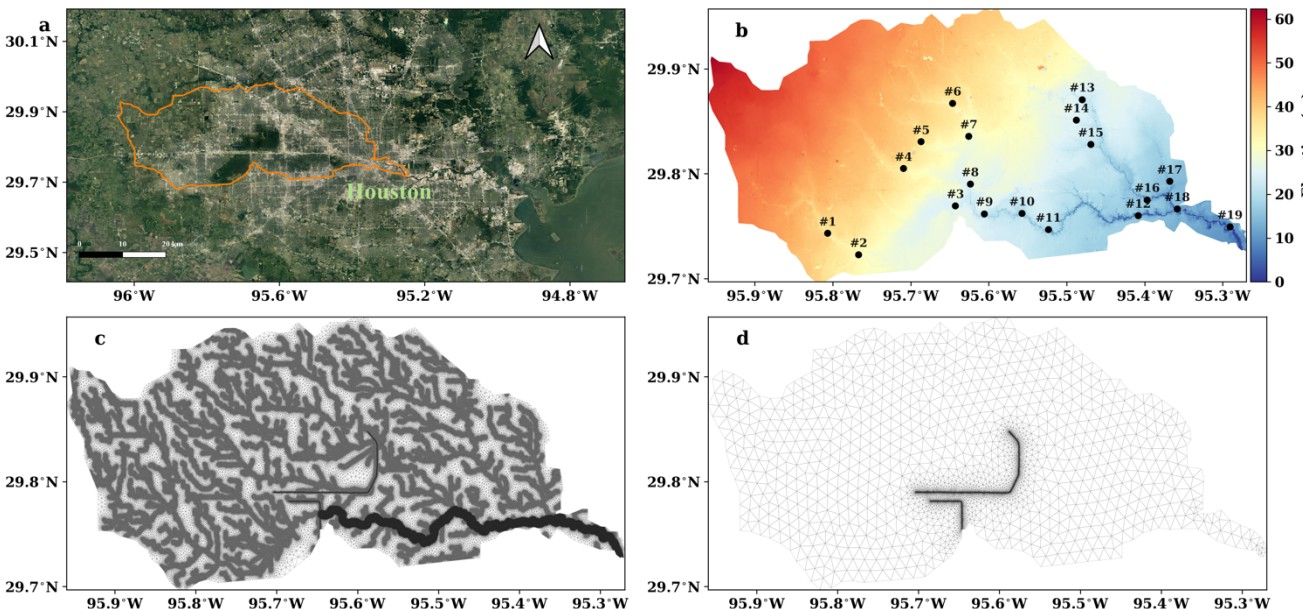

**Figure 2: Study domain (a), topography (b), high-resolution mesh (c), and coarse-resolution mesh (d) for river hydrodynamics
simulations in the Turning River basin. The river basin boundary is highlighted in orange color in (a) and the black dots in (b) show the
locations of the USGS gauges (Table S1) along with their gauge number. The basemap in (a) is extracted from © Google Imagery
@2024 TerraMetrics, Map data @2024. The seemingly thick bold lines in (c) are dense grid cells for river channels. The seemingly
black lines in (d) are dense grid cells for two reservoirs, which can also be seen in (c).**

Precipitation during Hurricane Harvey (Fig. 3) is extracted from the 1-km resolution Multi-Radar Multi-Sensor (MRMS)
precipitation dataset, which has a native temporal resolution of two minutes (Zhang et al., 2016). To demonstrate the
effectiveness of our downscaling approach for ensemble flood projections, we use a projected hurricane event (Hurricane
Harvey-like) under the high warming scenario — Shared Socioeconomic Pathway SSP5-8.5 — as a test case (Fig. 3). The
future hurricane is simulated using the Energy Exascale Earth System Model (E3SM) with the novel Simple Cloud-Resolving
E3SM Atmosphere Model (SCREAM) configuration (Caldwell et al., 2021; Donahue et al., 2024). SCREAM is a global
atmospheric circulation model with a non-hydrostatic dynamical core and parameterizations for atmospheric radiative transfer,
cloud microphysics, and boundary layer clouds and turbulence (Caldwell et al., 2021). The SCREAM domain features a
regionally refined mesh (RRM) with 3.25 km grid spacing over the east coast of the United States, including the Gulf of



Mexico and a significant part of the Atlantic Ocean, within a global domain that has 25 km grid spacing outside the RRM.
Nudging is applied to grid cells outside the RRM to constrain the atmospheric circulation using the European Centre for
Medium-Range Weather Forecasts (ECMWF) Reanalysis version 5 (ERA5) data (Hersbach et al., 2020). These features enable
SCREAM to capture fine-scale extreme weather events, accurately resolve coastal areas and mountainous regions, and
properly represent convective clouds, which are major sources of climate model uncertainty (Sherwood et al., 2014). SCREAM

is coupled with the E3SM Land Model (ELM), while sea surface temperature and sea ice extent are prescribed based on ERA5.

In the historical simulation, SCREAM is initialized using ERA5 to simulate Hurricane Harvey (hereafter referred to as
the SCREAM simulation). To simulate how Hurricane Harvey will behave under future conditions, a storyline simulation
using SCREAM is performed (hereafter referred to as the Pseudo Global Warming (PGW) simulation). In the PGW simulation,
the initial conditions and nudging data from ERA5 are perturbed by adding the mean monthly changes derived from a multi-

model ensemble of climate simulations from the Coupled Model Intercomparison Project Phase 6 (CMIP6) to represent the
mean climate change under the SSP5-8.5 scenario by the end of the 21st century (2079-2099) compared to the historical climate
at the end of the 20th century (1990-2010). A similar perturbation is also applied to ELM for the PGW simulations.

SCREAM can successfully predict the heavy precipitation during Harvey's first landfall, but its simulated precipitation
during the second landfall is relatively muted (Fig. 3a), a well-known challenge even for many weather forecasting models.

Considering the high computational cost of the SCREAM runs, we conducted three PGW simulations to drive the ensemble
flood projections. These simulations, each with slightly different initial conditions, represent the uncertainty of the hurricane
projection due to internal variability at the weather timescale (Fig. 3d). One PGW simulation is selected for LSG model
validation. Its temporal and spatial patterns of precipitation are shown in Figs. 3a and 3c, respectively, which are distinct from
the patterns of the benchmark precipitation (Figs. 3a and 3b) selected for LSG model training. The simulation differences

reflect model uncertainty and the effects of climate change on the hurricane. The distinct spatial pattern of precipitation in the
observations and the PGW simulation supports the latter as an out-of-sample test case, relevant for projecting future flooding.



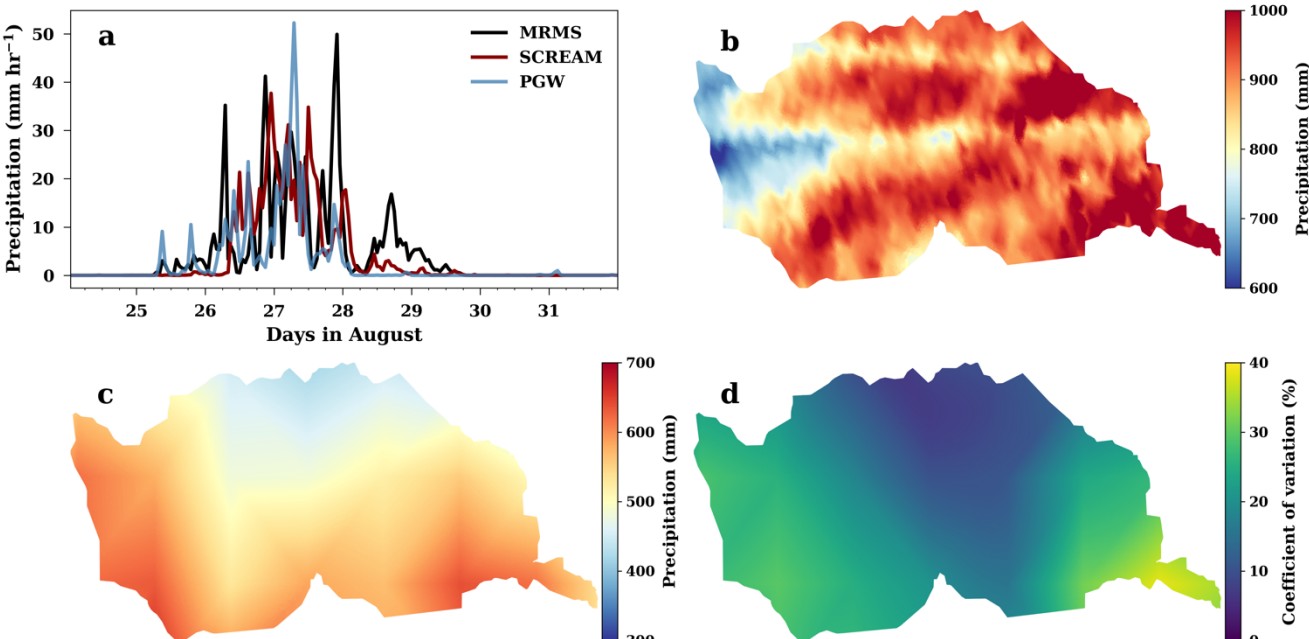

**Figure 3. Comparison of the observed (MRMS) and simulated hourly precipitation during Harvey under the observed historical (SCREAM) and projected future (PGW) conditions (a), and maps of the observed cumulative precipitation during Harvey (b), the cumulative precipitation of the PGW simulation selected for the LSG model validation (c), and the coefficient of variation (CV) of the PGW simulated cumulative precipitation ensemble (d) in the study domain.**

## 2.3 River Hydrodynamic Model

In this study, we chose the 2-D Overland Flow Model (OFM; Kim et al., 2012) for river hydrodynamics modeling, which was recently validated for the Harvey flood simulations (Xu et al., in review). In brief, OFM is a finite volume model that implements the first-order Godunov-type upwind scheme on a triangular mesh and uses Roe's approximate Riemann solver to compute fluxes between grid cells (Begnudelli & Sanders, 2006). Later, Ivanov et al. (2021) improved OFM's computational efficiency by using the Portable, Extensible Toolkit for Scientific Computation (PETSc; Balay et al., 2019) software for model parallelization. Mathematically, OFM solves the 2-D shallow water equations, which include the terms of advection, bottom friction, and gravity but ignore the terms of Coriolis and viscous forces (Begnudelli & Sanders, 2006):

$$\frac{\partial h}{\partial t} + \frac{\partial (uh)}{\partial x} + \frac{\partial (vh)}{\partial y} = q, \tag{4}$$

$$\frac{\partial (uh)}{\partial t} + \frac{\partial \left(u^2 h + \frac{1}{2}gh^2\right)}{\partial x} + \frac{\partial (uvh)}{\partial y} = -gh\frac{\partial z_b}{\partial x} - C_D u\sqrt{u^2 + v^2}, \tag{5}$$

$$\frac{\partial (vh)}{\partial t} + \frac{\partial (uvh)}{\partial x} + \frac{\partial \left(v^2 h + \frac{1}{2}gh^2\right)}{\partial y} = -gh\frac{\partial z_b}{\partial y} - C_D v\sqrt{u^2 + v^2}, \tag{6}$$

where $t$ is the time (s), $h$ is the flow depth (m), $u$ and $v$ are the water velocity (m s$^{-1}$) in the x and y direction under the Cartesian coordinate system, $q$ is the excess precipitation rate (m s$^{-1}$), $g$ is the gravitational acceleration constant (m s$^{-2}$), $z_b$ is the bed elevation (m), and $C_D$ is the bed drag coefficient derived from Manning's roughness $n$ as $C_D = gn^2 h^{-1/3}$.



We configure the OFM model on two variable-resolution meshes, with the high-resolution configuration serving as the HF RHM and the low-resolution configuration as the LF RHM. The variable-resolution meshes are generated using a Delaunay-based unstructured mesh generator, JIGSAW (Engwirda, 2017), which can refine topographic features important for shaping river flow regimes, such as river channels (Kim et al., 2022; Xu et al., 2022), floodplains (Yamazaki et al., 2011; Schrapffer et al., 2020), and water management structures (Schmutz & Moog, 2018). Specifically, the high-resolution mesh has 664,724 grid cells over the study domain, representing the main channels, tributaries, dams, and other regular cells with resolutions of 30 m, 60 m, 30 m, and 1000 m, respectively. In contrast, the low-resolution mesh has only 14,536 grid cells over the study domain, representing the main channels, tributaries, and other regular cells with a uniform resolution of 1000 m (except for dams, which are resolved at 30 m) (Fig. 2). In both the high-resolution and low-resolution meshes, the areas around two flood control reservoirs, Addicks and Barker's Reservoir (Fig. 2), are refined to ensure more accurate flood simulations. As indicated in Xu et al. (in review), even though the simulation of streamflow at the outlet is only moderately degraded, the use of a coarser mesh severely deteriorates the model performance in simulating inundation. The 30-meter resolution Digital Elevation Model (DEM) from the National Elevation Database (NED) was used to construct the topography of the RHM meshes.

To force the OFM, the MRMS precipitation data are upscaled from their native temporal resolution to an hourly time step and spatially interpolated to the variable-resolution mesh cells using the nearest neighbor interpolation method. Similarly, the hourly SCREAM simulation data are spatially interpolated to the variable-resolution mesh cells using the nearest neighbor method before being used to force the OFM.

## 3 Results

The trained LSG models can accurately predict the spatial and temporal variabilities of flow depth (Figs. 4–5) and velocity (Figs. 6–7) for the PGW flood event. First, the results confirm the effectiveness of the EOF analysis in extracting the significant spatial and temporal modes of the 2-D shallow water equations (Figs. S4–S5). Notably, as indicated by the proportion of variance explained by the specific modes, the significant modes of flow velocity (Fig. S5) are less representative of its variability compared to those of flow depth (Fig. S4), likely due to the higher nonlinearity of flow velocity simulations. Second, the trained LSG models perform remarkably well in reconstructing the HF ECs of river hydrodynamics from the LF ECs for both the training (Figs. S6–S7) and prediction phases (Figs. S8–S9). This performance is achieved despite substantial distinctions between the HF and LF ECs. Consequently, the spatial and temporal features of the high-resolution flow depth and velocity are well reproduced for both the training (Figs. S10–S13) and prediction phases (Figs. 4–7), even though they are forced by two distinct hurricane events (Fig. 3).






**Figure 4. Root-mean-square-error (RMSE) of the LF simulated (a) and downscaled flow depth (b) for the PGW event, the LF simulated (c) and downscaled flow depth (d) at 01:00 am, August 27, the bias of the LF simulated (e) and downscaled flow depth (f) at the same time, and the HF simulated flow depth (g) at the same time. RMSE and bias are calculated by treating the HF simulation as "ground truth".**

During the PGW flood, the average root-mean-square-error (RMSE) of the downscaled flow depth is 0.07±0.1 m (Fig. 4b), which is less than one-fourth of the average RMSE (0.3±0.6 m) of the simulated LF flow depth (Fig. 4a). The downscaling achieves impressive error reductions in river channels (particularly downstream reaches), the nearby floodplains, and the two reservoirs (Fig. 4), which are flood-prone areas that have been deliberately refined in the high-resolution mesh (Fig. 2c). By downscaling, the detailed longitudinal variations of flow depth are precisely reproduced (Fig. 4d) during the peak flood period

(near 01:00 am, August 27). Even very small ponding grid cells, which are barely seen in the LF simulation (Fig. 4c), are recovered (Fig. 4d). Compared to the LF simulation, the downscaled flow depth is highly consistent with the HF simulation, with the bias range (from the 10th percentile to the 90th percentile) reduced from [-0.2 m, 0.3 m] (Fig. 4e) to [-0.04 m, 0.06



m] (Fig. 4f). Generally, the downscaling reduces the underestimation and overestimation of the LF simulated flow depth in
river channels and floodplains, respectively, likely due to the use of the HF EOFs (Fig. S4).

**Figure 5. Comparison of the HF simulated, LF simulated, and downscaled flow depth at the selected USGS gauges during the PGW event.**

The downscaling approach also performs promisingly in reproducing the temporal variability of the HF simulated flow depth at the selected USGS gauges (Fig. 5). Assessed by the Kling-Gupta efficiency (KGE) (Gupta et al., 2009), the downscaled flow depth shows good performance (KGE ≥ 0.5) at all gauges except Gauge #14, where small inundation occurs. In contrast, the LF simulation only shows good performance at two gauges but poor performance (KGE < -0.41; Knoben et al., 2019) at three gauges, whereas the performance at the other gauges is barely acceptable (-0.41 < KGE < 0.5). Notably, for four gauges (#3, #7, #8, and #19), the performance of the downscaling approach is excellent (KGE ≥ 0.9). Not only does the downscaling reduce the severe biases of the LF simulation at nearly all gauges, but it also recovers dynamics not captured by the LF simulation, such as the second peak flow depth at Gauge #18.





**Figure 6. RMSE of the LF simulated (a) and downscaled flow velocity (b) for the PGW event, the LF simulated (c) and downscaled flow velocity (d) at 01:00 am, August 27, the bias of the LF simulated (e) and downscaled flow velocity (f) at the timestep, and the HF simulated flow velocity (g) at the timestep. RMSE and bias are calculated by treating the HF simulation as "ground truth".**

Likewise, the downscaled simulations provide accurate representations of the spatial and temporal variabilities of flow velocity during the PGW flood (Figs. 6–7). The downscaling significantly reduces the average RMSE of simulated flow velocity from $0.7\pm1.9$ m s$^{-1}$ (Fig. 6a) to $0.2\pm0.6$ m s$^{-1}$ (Fig. 6b). Compared to flow depth, the error reduction in flow velocity is more concentrated in the river channels, possibly reflecting the larger gradients of flow velocity from river channels to the nearby floodplains. Like flow depth, the downscaling successfully recovers the detailed longitudinal variations of flow

velocity, as well as the river flow in small inundated areas during the peak flood period (Fig. 6d). The method also yields substantial reductions in the estimation bias, from [-0.4 m s$^{-1}$, 0.3 m s$^{-1}$] (Fig. 6e) to [-0.07 m s$^{-1}$, 0.05 m s$^{-1}$] (Fig. 6f).





**Figure 7. Comparison of the HF simulated, LF simulated, and downscaled flow velocity at the selected USGS gauges during the PGW event.**

The downscaling also produces more consistent temporal variability of flow velocity compared with the HF simulation at the selected USGS gauges (Fig. 7). The downscaled flow velocity demonstrates good performance at 15 gauges and excellent performance at 2 gauges (#11 and #16). In contrast, the LF simulation only shows good performance at Gauge #8 but exhibits unacceptable performance at 9 gauges (e.g., KGE < -0.41). However, despite its superiority to the LF simulation, the downscaled flow velocity does not perform as well as the downscaled flow depth at many of the study gauges. For instance, it

fails to capture the velocity spike at Gauge #1 and greatly underestimates the velocity peaks at several other gauges (e.g., #2, #4, and #8). The downscaled solutions also struggle to reproduce the high-frequency fluctuations of flow velocity, such as at Gauges #12 and #18. Analysis of the error sources indicates that for the downscaled flow velocity, the error from dimensionality reduction ERDR is substantially larger than the LSG model error ERLSG, while for the downscaled flow depth, ERLSG is dominant (Fig. 8). First, this result aligns with the EOF analysis (Fig. S5), which shows the higher nonlinearity of

flow velocity simulations. Second, this implies that reducing ERDR is crucial for more accurate flow velocity downscaling.





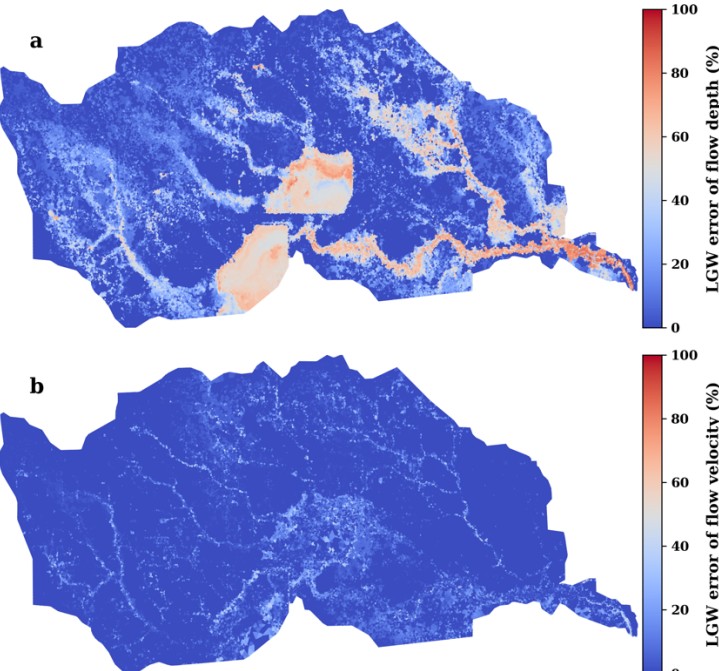

**Figure 8. Percentage of flow depth (a) and velocity (b) downscaling uncertainty that can be explained by the error from the LSG model.**

A possible way to reduce ERDR is to regionalize the training of the LSG model in a smaller domain that focuses on a
specific geographic feature. This approach can prevent locally important EC modes from being filtered out in large-scale EOF analyses (see Fraehr et al. (2023a) for North's test and Kaiser's Rule). Notably, this treatment does not require new model simulations and follows the same procedure outlined in Fig. 1. We selected Gauge #1, where the whole-domain downscaling fails to reproduce the peak flow velocity simulated by the HF model. By training a new LSG model over a smaller area encompassing the gauge (Fig. S14), the downscaled simulation aligns with the HF model for predicting the flow velocity spike
on August 26 (Fig. 9). For the PGW event, the KGE for the simulated flow velocity increases significantly from 0.04 to 0.61. The regionalized training also slightly improves the accuracy of the downscaled flow depth at Gauge #1, with KGE increasing from 0.81 to 0.96. The smaller effect of regionalized training on flow depth is expected because our error analysis indicates that the uncertainty of the downscaled flow depth is only minorly contributed by ERDR (Fig. 8).



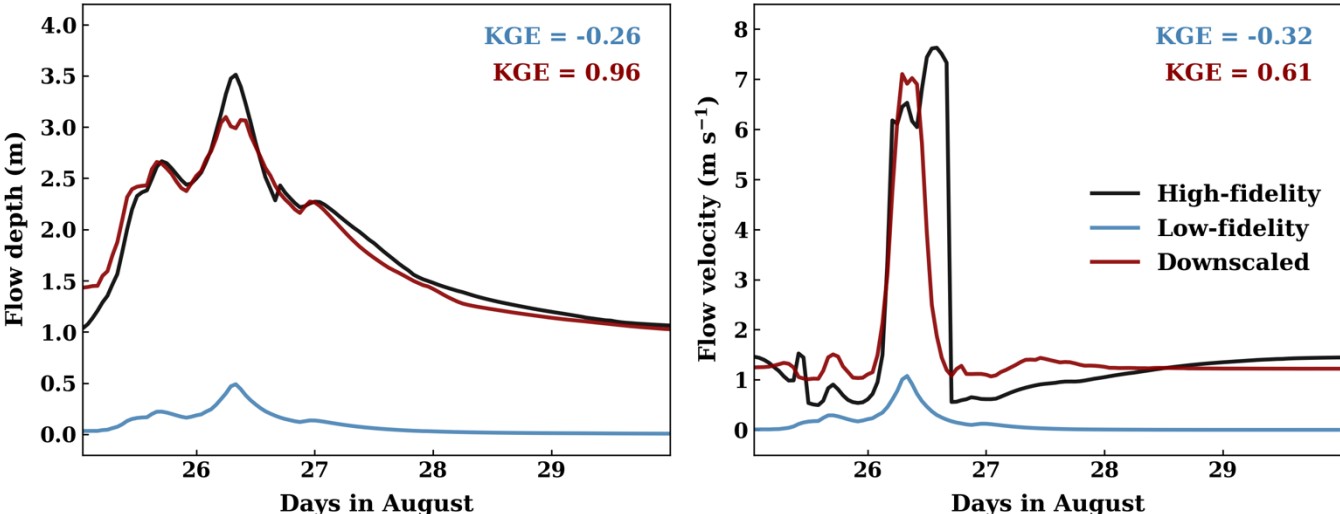

**Figure 9. Comparison of the HF simulated, LF simulated, and downscaled flow depth and velocity at Gauge #1 during the PGW event with training LSG models in a focused area around the gauge (Fig. S14).**

    Because the training of the LSG model can be completed within minutes, the computational cost of our downscaling approach depends solely on the computational time needed for the RHM simulations. For the 13-day PGW simulations, when running on Intel Xeon Skylake CPUs (2.4 GHz) with 192 GB of DDR4 DRAM, the HF model (664,724 grid cells) requires

4,032 CPU hours to complete, while the LF model (14,536 grid cells) requires only 48 CPU hours. Thus, by applying the downscaling approach to the LF ensemble simulations, our method provides an efficient way to evaluate the impact of the uncertainty in tropical cyclone (TC) predictions on the simulation of urban flooding. Figure 10 shows that compared to the single-member PGW simulation described in the above evaluation, a three-member ensemble of PGW simulations predicts higher peak inundation depths in the lower reaches of the Buffalo Bayou watershed, where population density is also the

highest. In some areas, the difference in peak flood depth during the PGW event can exceed 1 m (Fig. 10b). Using the ensemble simulations, we can also calculate the likelihood of the areas where the PGW flood event poses significant or high risks to human safety ($h > 0.95$ m; Russo et al., 2013). From the ensemble simulations, humans will very likely face significant risks to the flood water in the two reservoirs, river channels, and the nearby areas in Houston during the PGW flood event (Fig. 10c). In line with Fig. (10b), some simulations predict larger extents of the areas where the flood event would pose significant risks

to human safety.



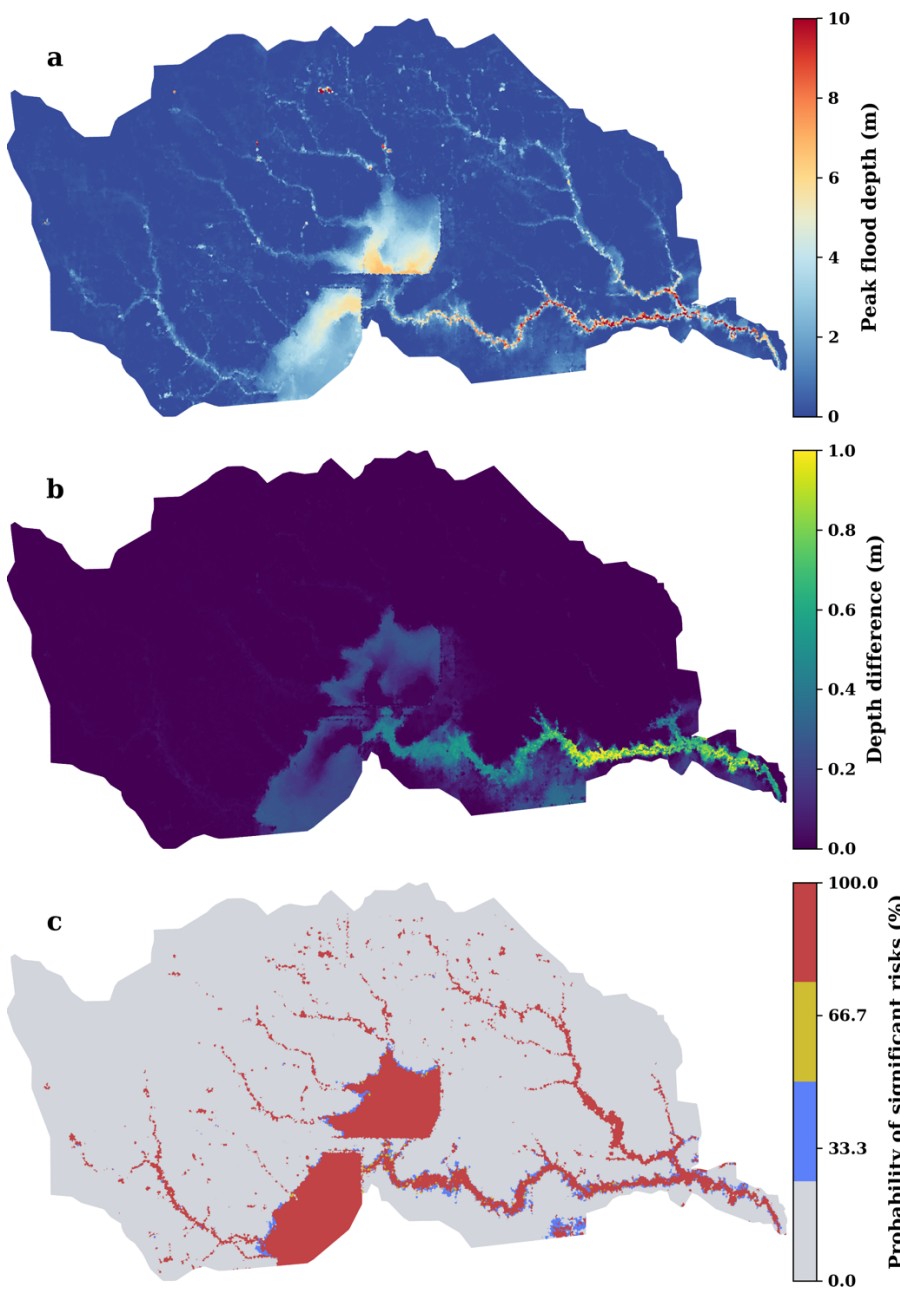

**Figure 10. Projected peak flood depth of the PGW ensemble simulation (a), the difference of the projected peak flood depth between the PGW ensemble simulation and the selected PGW simulation (b), and the probability of significant risks to human safety by flood (c).**





## 4 Discussions

### 4.1 Simulation of high-resolution river hydrodynamics

Our results demonstrate that the LSG model-based downscaling approach can provide efficient and accurate simulations of high-resolution river hydrodynamics at the computational cost of LF RHMs. To the best of our knowledge, this is one of the first studies to explore methods for fast and accurate simulations of high-resolution flow velocity in realistic cases, broadening the usefulness and relevance of recent rapid progress in hydrodynamic modeling, which still exclusively focuses on flooding (Carreau & Guinot, 2021; Xie et al., 2021; Zhou et al., 2021; Feng et al., 2023b; Fraehr et al., 2023b; Frame et al., 2024; Wing et al., 2024). With HF simulations of flow velocity, our understanding of not only instantaneous flood hazards but also longer time-scale environmental hazards, such as eutrophication and pollution, can be greatly advanced. More broadly, the new method can contribute to the development of fully coupled atmosphere-land-river-ocean ESMs, which will be discussed in detail in Section 4.2. It is worth noting that the study watersheds of Fraehr et al. (2023a, 2023b) differ from this study in land use and climate. The two Australian watersheds in Fraehr et al. (2023a, 2023b) are dominated by rural and natural landscapes and are less affected by TCs. The success of the LSG model in different domains underscores its broad geographical applicability.

The LSG model-based downscaling approach has two major advantages over neural network (NN)-based methods for high-resolution river hydrodynamic modeling. First, compared to NN-based methods (Tran et al., 2023), the training time of the LSG method is negligible, requiring only one expensive HF RHM simulation for training. Second, because physical laws have been explicitly coded in LF RHMs and implicitly complied with in the spatial interpolation process, the trained model can be expected to be transferable to future unseen climate conditions. These advantages make the approach well-suited for ensemble projections of future flooding, which are crucial for robust assessment of flood adaptation and mitigation (Fig. 10) given the substantial uncertainty of TC projections (Fig. 3). Another potential strength of our approach is that it can directly benefit from future advances in RHMs. The development of better RHMs will provide more accurate LF and HF simulations of river hydrodynamics for LSG model training, helping to reduce downscaling uncertainty (Fraehr et al., 2023a).

While a well-trained LSG model can be applied to unseen climate conditions, it is not free from re-training. For instance, without re-training, an LSG model is unlikely to handle changes in land use and geographical features, such as geomorphological changes in river channels and river flow modifications related to reservoirs. Additionally, our training strategy, which trains the LSG model only with data from the Harvey flood event, may not be effective in more complex cases where floods are not always driven by TCs. For instance, the main flood mechanisms in the U.S. Mid-Atlantic watersheds include both rain-on-snow (ROS) and snowmelt events that mainly occur in high-latitude areas (e.g., 1996 ROS Flood), and heavy rainfall from tropical cyclones (e.g., Hurricane Irene in 2011), extratropical systems, and convective systems (Smith et al., 2010; Li et al., 2021; Sun et al., 2024). For such cases, it is necessary to follow the training procedure of Fraehr et al. (2023a), selecting multiple representative flood events of different types for training. Since the number of flood mechanisms




is limited, we expect that the computational demand will still be manageable even if the LSG model is applied to a watershed with diverse flood generation processes.

Our study reveals that the downscaling accuracy of flow velocity is lower than that of flow depth. This is because the dynamics of flow velocity are more nonlinear, which induces significantly larger dimensionality-reduction errors in the downscaling process (Fig. 8). Accordingly, we introduced a regionalized training procedure to improve the downscaled flow velocity in focused areas (Fig. 9). This procedure does not significantly increase the computational cost of the LSG model because it does not require any new RHM runs. We envision that this strategy can be particularly useful for simulating river hydrodynamics in geographical areas that need more careful flood risk assessments, such as schools, hospitals, critical

infrastructures, energy facilities, and Superfund sites (Brand et al., 2018).

The LSG model error $ER_{LSG}$ primarily depends on the performance of the Sparse GP model in mapping LF ECs to HF ECs. Besides the Sparse GP model, other data-driven models, such as Multilayer Perceptrons and Artificial Neural Networks, can also be used to establish the complex relationships between ECs (Carreau & Guinot, 2021). Future research on implementing other data-driven models to reduce $ER_{LSG}$ is also worth exploring.

The LF model used in this study is about 84 times faster than the HF model, which is more efficient than the LF model adopted by Fraehr et al. (2023a) and achieves a larger acceleration rate than the theoretical boost rate when considering the reduction in the number of grid cells ($\frac{664,724}{14,536} \approx 46$). The improved efficiency indicates that the OFM RHM has taken advantage of fewer computational units and longer time steps according to the Courant–Friedrichs–Lewy convergence criteria in the simulations. Furthermore, the results underscore the usefulness of our approach for flood risk assessment which needs hundreds

or thousands of ensemble model runs (Wu et al., 2020).

## 4.2 Coupling large-scale hydrodynamical processes with local processes in river models

It is challenging to represent other physical, chemical, and biological processes beyond river discharge in large-scale river models. This is mainly because, by sacrificing process and resolution accuracy for computational efficiency, these models cannot provide accurate simulations of high-resolution flow depth and velocity necessary for calculating local dynamics

important for fluvial processes (Bertagni et al., 2024), such as sediment settling velocity (Li et al., 2022), bottom shear stress and diffusivity (Chen et al., 2023), and greenhouse gas outgassing velocity (Ulseth et al., 2019). By accurately and efficiently simulating high-resolution flow depth and velocity, our downscaling approach provides an opportunity to bridge the gaps between large-scale hydrodynamical processes and detailed local processes in river models. Specifically, we propose a two-way coupling scheme in large-scale river models (Fig. 11). In the first stage of each simulation cycle, a large-scale river model

is used to simulate coarse-resolution flow depth and velocity and transport mass and momentum downstream. In the second stage, the LSG model-based approach is employed to downscale the simulated flow depth and velocity to fine resolutions. In the third stage, high-resolution hydrodynamics are used to drive detailed physical, chemical, and biological models, such as the PFLOTRAN model for geochemistry (Hammond et al., 2012) and the GAIA model for sediment (Tassi et al., 2023), to





simulate the sources and sinks of the represented tracers. In the final stage, the sources and sinks calculated in the fine-
resolution mesh are upscaled to the coarse-resolution mesh of the large-scale river model and used to update the concentrations
of the represented tracers.

An outstanding weakness of existing ESMs is that they ignore the lateral biogeochemical fluxes in the land-river-ocean
continuum and therefore do not close the global biogeochemical cycles (Regnier et al., 2022). By implementing this new
paradigm of river modeling in ESMs, land, river, and ocean biogeochemistry will be fully coupled, helping to close the global
biogeochemical cycles. To achieve this vision, future research must focus on extending the LSG model-based approach to
downscale 1-D river models to 2-D fine-resolution meshes. This is because, despite the prospect of 2-D large-scale river
models running on GPU-based supercomputers, 1-D river models will likely still be the default configuration in ESMs in the
near future (Telteu et al., 2021). We envision that the potential challenges could include the alignment of 1-D and 2-D
unstructured meshes and the interpolation of simulated 1-D river hydrodynamics onto 2-D meshes.

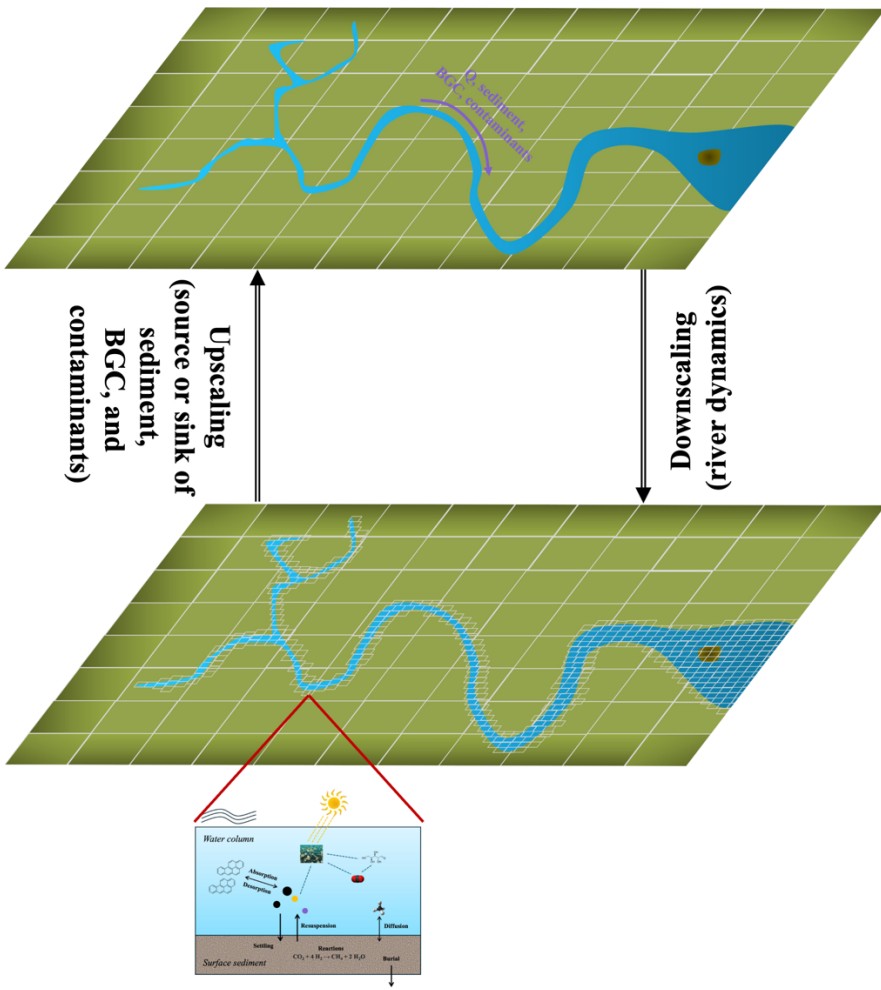


**Figure 11. A schematic illustration of coupling large-scale hydrodynamical processes that are simulated in coarse resolution with local physical, chemical, and biological processes that are simulated in fine resolution in river models.**



## 5 Conclusion

In this study, we developed a downscaling approach based on the LSG model to achieve fast and accurate simulations of
high-resolution river flow depth and velocity. Our test of TC-induced flood events in an urban watershed in Houston
demonstrates the effectiveness and efficiency of the downscaling method, as the simulation errors in the LF RHM are greatly
reduced, without additional computational costs. We further indicated that the simulation error of the downscaled flow velocity
can be reduced by employing regionalized training of the LSG model for selected focused areas. As one of the first studies to
explore high-fidelity and efficient flow velocity simulations in realistic cases, our research can help broaden the usefulness
and relevance of the recent rapid progress in hydrodynamic modeling, which still exclusively focuses on flooding. More
importantly, the downscaling approach provides an opportunity to bridge the gaps between large-scale hydrodynamical
processes and local physical, chemical, and biological processes in river models, which could eventually help close the global
biogeochemical cycles in ESMs.

*Code and data availability*. The code and input data for this work can be publicly available at
https://doi.org/10.5281/zenodo.14258083.

*Acknowledgements*. This study was supported by the U.S. Department of Energy Advanced Scientific Computing Research
(ASCR) program through the Multiphysics Simulations and Knowledge discovery through AI/ML technologies (MuSiKAL)
project. It was also partly supported by the Scientific Discovery through Advanced Computing 5 (Capturing the Dynamics of
Compound Flooding in E3SM) and the Integrated Coastal Modeling (ICoM) project, funded by the U.S. Department of Energy,
Office of Science, Office of Biological and Environmental Research as part of the Earth System Model Development (ESMD)
and Regional and Global Model Analysis (RGMA) program areas, respectively. The Pacific Northwest National Laboratory
(PNNL) is operated by Battelle for the U.S. Department of Energy under Contract DE-AC05-76RLO1830. All model
simulations were performed using resources available through (a) Research Computing at PNNL and (b) the National Energy
Research Scientific Computing Center (NERSC), a U.S. Department of Energy Office of Science User Facility located at
Lawrence Berkeley National Laboratory, operated under Contract No. DE-AC02-05CH11231 using NERSC award BER-
ERCAP0027117.

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
