# Peer review of "An efficient hybrid downscaling framework to estimate highresolution river hydrodynamics"

_EGUsphere, 2024_

## Referee Comment (RC3)

[referee-annotated manuscript omitted]

---

## Author Response (AR1)

**Response letter to Reviewer #1**

**The manuscript presents a hybrid downscaling framework aimed at improving the accuracy and computational efficiency of high-resolution river hydrodynamics, including both flow depth and velocity. The authors used an LSG model developed by Fraehr et al. to construct high-resolution flow depth and velocity from a low-fidelity 2-D RHM simulation. The paper is generally well written and the results are clearly presented. The computational efficiency demonstrated (84-fold speed-up) is impressive. Here are some specific comments.**

Response: We thank the reviewer for the valuable comments. Our point-to-point responses are provided below. The revisions are highlighted in the revised manuscript with tracked changes. The listed line numbers are based on the clean version.

**(1) The paper presents a nice follow-up study of Fraehr et al. (2022 & 2023a). Unlike Fraehr's previous studies, flow velocity was also downscaled. The training of the Sparse GP models was performed independently for flow depth and velocity. However, the difference between the characteristics of depth and velocity is not very clear to me. It looks like the method is the same as Fraehr's papers except for a few details. The authors stated that "this is one of the first studies to explore methods for fast and accurate simulations of high-resolution flow velocity". Is it simply because the training of velocity models is so difficult that nobody is doing it? My suggestion is to focus on flow velocity, and perhaps incorporate a comparison between the LSG model and other velocity downscaling models in terms of both accuracy and computational cost, which would help place this work in a broader context and demonstrate its relative advantages.**

Response: We thank the reviewer for the comments, which make us recognize the need to more clearly differentiate our work from Fraehr et al. (2022 & 2023a) because our study is not simply a follow-up study of Fraehr et al. (2022 & 2023a). Except for extending the downscaling method developed by Fraehr et al. (2022 & 2023a), our study is unique in many aspects, including the use of a full-dynamic 2-D hydrodynamic model, the focus on a highly urbanized watershed where flooding is mainly driven by hurricanes, and leveraging a state-of-the-art atmospheric model for future hurricane projections. More importantly, our study differentiates from Fraehr et al. (2022 & 2023a) in research objectives. Whereas Fraehr et al. (2022 & 2023a) solely focused on efficient and accurate flood simulations, our study aims to achieve efficient and accurate river

hydrodynamic simulations that can improve Earth system modeling. This difference also explains why our study includes the downscaling of river velocity, which has been overlooked in existing studies.

To better illustrate the motivation and uniqueness of our study, we have revised the manuscript (Lines 79-92) as "However, like Fraehr et al. (2022 & 2023a), existing research on hydrodynamic model downscaling is mostly flood prediction oriented and thus has focused entirely on flood extent and magnitude, while ignoring flow velocity. This overlook is problematic from two perspectives. First, it could increase the uncertainty of flood risk simulations because flood velocity is a critical factor for human safety risks in flood events (Russo et al., 2013). Moreover, as discussed earlier, in the context of Earth system modeling, without accurate simulations of flow velocity, it is not possible to realistically predict how river functions respond to environmental stresses. While Fraehr et al. (2022 & 2023a) only applied the LSG-based downscaling approach for inundation extent and depth, this method should theoretically also be applicable for flow velocity. This is because the mass and momentum of river flow are governed by the unified shallow water equations and driven by the same environmental factors. However, such an application has not yet been explored.

In this study, we develop an LSG-based downscaling approach to achieve accurate simulations of high-resolution river flow depth and velocity at the computational cost of a low-resolution 2-D RHM. Compared to Fraehr et al. (2022 & 2023a), the main innovation of our study is to test and enhance the LSG-based downscaling approach for high-resolution flow velocity downscaling. This extension of the LSG-based downscaling approach is expected to greatly broaden its usefulness for Earth system modeling."

**(2) The model was validated for a single flood event in Houston. The authors stated "The effectiveness and transferability of our method are tested in an urbanized watershed in the Houston area using data from two extreme hurricane events". How could one flood prove its transferability? The incorporation of more historical events and observed hydroclimatic data could strengthen the claim of model generalizability.**

Response: Sorry for the confusion. We should state clearly that the approach's transferability would be demonstrated if its effectiveness can be validated in both Fraehr et al. (2022 & 2023a) and our independent study. To avoid such confusion, we have revised the sentence (Lines 92-95)

as "Besides, we test the effectiveness of the downscaling method in an urbanized watershed in the Houston area using data from two different extreme hurricane events. Together with Fraehr et al. (2022 & 2023a), our independent validation in a different environment would help examine whether the LSG-based downscaling approach has broad geographical applicability."

**(3) "Furthermore, based on this downscaling method, we propose a new paradigm to couple large-scale hydrodynamical processes with local detailed physical, chemical, and biological processes in river models". This is exciting but not validated in the paper.**

Response: We thank the reviewer for acknowledging the value of this new paradigm. Although it would be exciting to validate the idea, we think that it would not be a good idea to explore and demonstrate the feasibility of this idea within the current study in a single manuscript. First, the objectives of the current work are not only the introduction of the new paradigm but also the demonstration of the effectiveness of the downscaling approach for ensemble flood simulations. Second, the research included in the current work, particularly the extension of the downscaling approach to flow velocity, deserves a dedicated manuscript to describe all nuances. Third, the validation of the downscaling approach is only the first step in realizing the new paradigm, which would need a substantial amount of additional work. Moreover, we introduced the new paradigm as a vision instead of a product to motivate the scientific community for its realization.

**Response letter to Reviewer #2**

**The study presented a good downscaling method for hydrodynamics from low-quality outputs to high-quality outputs. I think the method is innovative and promising since it can save considerable computational time. The paper is well written. I recommend the journal to publish this study, but I think some minor improvement is needed.**

Response: We thank the reviewer for the valuable comments and for acknowledging the importance of our study. Our point-to-point responses are provided below. The revisions are highlighted in the revised manuscript with tracked changes. The listed line numbers are based on the clean version.

**1) The author should clarify what the main innovation that is presented by the authors from that in Fraehr et al. (2022).**

Response: Thanks for the comment. In the revision, we clarify the main innovation of our study from that in Fraehr et al. (2022, 2023a) as "Compared to Fraehr et al. (2022 & 2023a), the main innovation of our study is to test and enhance the LSG-based downscaling approach for high-resolution flow velocity downscaling" in the introduction (Lines 89-91). Also, we highlight one advantage of our approach relative to Fraehr et al. (2022 & 2023a) as "The LF model used in this study is about 84 times faster than the HF model, which is significantly more efficient than the LF model adopted by Fraehr et al. (2023a) that is only 12 times faster. Also, our LF model achieves a larger acceleration rate than the theoretical boost rate when considering the reduction in the number of grid cells ($\frac{664,724}{14,536} \approx 46$). The improved efficiency indicates that the OFM RHM has taken advantage of fewer computational units and longer time steps according to the Courant–Friedrichs–Lewy convergence criteria in the simulations. Furthermore, the results underscore the usefulness of our approach for flood risk assessment, which needs hundreds or thousands of ensemble model runs for uncertainty quantification (Wu et al., 2020), for which the configuration of Fraehr et al. (2023a) cannot provide due to its inefficient LF simulations" in the discussion section (Lines 493-500).

**2) In the result section, I recommend the authors should add some capitals for subsections, since there are many figures listed.**

Response: Thanks for the comment. In the revision, we have added three subsections in the results section.

**3) I do not agree to the last two comments that were posted by the first reviewer. The reviewer think that one single flood event is not sufficient to prove transferability of the method. I do not think so, because the numerical simulation of a flood actually need numerous time steps, as long as in the first hundreds of time steps can perform well, the method will inevitably perform well for further more time steps. There is no need to do with other more flood events. The first reviewer also think that the authors "propose a new paradigm to couple large-scale hydrodynamical processes with local detailed physical, chemical, and biological processes in river models", but not validated in the paper. I think this new paradigm is presented in the discussion section. This theoretical assumption is interesting and promising. I do not think such a discussion must be validated in the current study.**

Response: Thanks for acknowledging the validity and contribution of our study. We fully agree with the reviewer's arguments.

**Response letter to Reviewer #3**

**I reviewed the manuscript entitled "An efficient hybrid downscaling framework to estimate high-resolution river hydrodynamics" by Tan et al. The manuscript applies the Fraehr et al. (2022) method but for the simulated flow depth and velocity, which is different than Fraehr et al. (2022) that is on the flood extent. The manuscript assesses the utility of the method using Hurricane Harvey. Overall, I think the manuscript is a novel one compared to Fraehr et al. (2022) given the different focusing variables. I only have one major concern and two suggestions on writing of the manuscript. There are also 22 minor comments/editing in the annotated manuscript. Therefore, I would suggest a moderate revision.**

Response: We thank the reviewer for the valuable comments and for acknowledging the importance of our study. Our point-to-point responses are provided below. The revisions are highlighted in the revised manuscript with tracked changes. The listed line numbers are based on the clean version.

**Major concern: The method is applied to only one event**

**I understand the HF RHM simulation is very time consuming. Instead of doing another case, I wonder is it possible to compare the results to some ground-based records and satellite observation? Harvey is a devastating event and I believe there should be plenty of measurements or reports on the inundation during the event. It would be great to see how are the LF, HF, and downscaled depths and velocities (hopefully have) compared to the ground-based observations.**

Response: Thanks for the comment. While we appreciate the reviewer's suggestion to compare our results with some observations, our goal is to develop and validate the downscaling approach, not the river hydrodynamic model. Hence, we design our experiments to include both high- (HF) and low-fidelity (LF) river hydrodynamic model (RHM) simulations, with the LF simulation providing the input data for downscaling and the HF simulation providing the "truth" or reference to evaluate the downscaled data. That is, downscaling aims to provide a mapping between the LF and HF simulations, not between the LF simulation and observations. Good agreement between the downscaled simulation and the HF simulation is a demonstration of the success of the downscaling method. This "perfect prognosis" approach to evaluating downscaling methods has been used in climate downscaling; it allows one to focus on evaluating the downscaling methods

without the influence of model or observation errors (Denis et al. 2002). In the new subsection 2.4, we have explained this validation strategy as "Validation of the downscaling method uses the "perfect prognosis" approach in which the HF RHM simulation is the target for the downscaled flow depth and velocity that uses the LF RHM simulated flow depth and velocity as the input. This validation strategy allows one to focus on evaluating the downscaling method without the influence of the RHM or observation errors and has been widely adopted in climate downscaling (Denis et al. 2002) as well as hydrological and hydrodynamic downscaling (Carreau & Guinot, 2021; Feng et al., 2023b) when both low- and high-resolution simulations are available. Therefore, in this study, good agreement between the downscaled flow depth and velocity with the high-resolution simulations from the HF RHM is a demonstration of the effectiveness of the downscaling method."

Even though the fidelity of the HF RHM is irrelevant in the perfect prognosis framework, validation of the HF RHM simulation in our study domain has been performed in our recently published WRR paper (Xu et al., 2025). The information has been provided in the revision. It should be noted that the HF RHM simulated flow velocity is not validated in Xu et al. (2025) because flow velocity measurements are not available at the selected USGS gauges during the simulation period.

We further note that the downscaling method has been evaluated using two case studies: Hurricane Harvey as it occurred in 2017 and a constructed Hurricane Harvey that could happen in the future, as provided by the PGW simulation. Due to global warming, the "future" Hurricane Harvey has distinct characteristics different from the Hurricane Harvey that occurred in 2017, making it an out-of-sample test case. In the revision, we have also updated Figure 3 to provide more evidence to support the use of the PGW flood event as an out-of-sample test case.

**Suggestions on improving the presentation of the manuscript**

**I have two suggestions regarding the presentation of the manuscript. First, I think the authors need to dedicate a subsection within the current Section 2 to introduce their strategies on evaluation of the model simulations. What is the reference? Which error metrics are used? What are the methods for validation? Currently, this information was given in different locations within Section 3, which is not good. Second, I think it is better to have subsections for the results section. Perhaps Figure 4 to 7 can be grouped into one**

**subsection, presenting the validation results; and Figure 8 to 10 is another for the uncertainty analysis.**

Response: Thanks for the comment. In the revision, we have added a new subsection 2.4 to describe our validation strategy and the evaluation metrics. Also, we have followed the suggestion to add subsections for the results section.

**Minor comments:**

**L11: process**

Response: Revised as suggested.

**L12: remove "accurately"**

Response: Revised as suggested.

**L12: I would suggest to provide the exact resolution. Same for the coarse resolution.**

Response: Thanks for the comment. Because this sentence is to provide general information about our methods, it may be not a good idea to specify the exact resolutions here. Instead, we have added a new sentence in the abstract (Lines 13-15) as: "The high-resolution (as fine as 30 m resolution) and low-resolution (mostly 1000 m resolution) meshes include 664,724 and 14,536 grid cells, respectively."

**Figure 2: Perhaps also show the boundaries of the reservoirs on panel a.**

Response: The boundaries of the reservoirs are added and highlighted in blue on panel (a).

**L173: Pseudo Global Warming, PGW, simulation**

Response: Revised as suggested.

**L178-179: Where are the first and the second landfalls? Visually, I can identified four landfalls.**

Response: Sorry for the confusion. Hurricane Harvey did make more than two landfalls. In the revision, we have clarified it as "SCREAM can successfully predict the heavy precipitation during

Harvey's landfalls in Texas on August 26, but its simulated precipitation during the subsequent landfalls on August 27 and 28 is relatively muted".

**L179: Is this specific to the prediction of Harvey or a general issue of any multi-modal storm event? Maybe adding some citations at the end of the sentence?**

Response: This is a well-known challenge specific to the prediction of Harvey. We have added two references (Wang et al., 2018; Yang et al., 2019) at the end of the sentence.

**Figure 4: I think it is better to also have it mentioned here that this is the peak time.**

Response: Thanks for the comment. We have added the information in both Figures 4 and 6.

**Figure 4: I think the authors need to dedicate a subsection in Section 2 to describe their strategies for performance evaluation.**

Response: Thanks for the comment. As mentioned above, we have added a new subsection 2.4 in the revision to describe the validation strategy.

**Figure 4: 1:00 am? I wonder why not showing the mean bias of the entire event.**

Response: It should be noted that the errors of the entire event have been evaluated using RMSE and shown in panels (a) and (b). Here, the bias is used for the peak flood only for two reasons. First, the peak flood is widely used for flood risk analysis and warning. Second, due to the potential time lag of the simulation and ground truth, the positive and negative biases can be canceled out in calculating the mean bias, leading to underestimated model errors.

**Figure 4: This panel was not discussed in the texts.**

Response: Panel g is mentioned in Line 309. Also, for Figure 6, panel g is mentioned in Line 347.

**Figure 6: I wonder how is the performance of velocity during low-flow condition. Is the modeling stable?**

Response: As shown in Figure 7, the model performs reasonably well during low-flow conditions.

**L271: Bias of the particular hour, right? The way you write is a bit confusing. It is not the overall bias.**

Response: Thanks for pointing out the confusion. In the revision, we have revised the sentence as "The method also yields substantial reductions in the estimation bias at the peak flood time" (Lines 347-348).

**L276: I would suggest to refer directly to the statistics instead of using adjective.**

Response: As suggested, we have referred directly to the statistics (Lines 389-391) as "The downscaled flow velocity achieves KGE ≥ 0.5 (good performance) at 15 gauges and KGE ≥ 0.9 (excellent performance) at two gauges (#11 and #16). In contrast, the LF simulation only achieves KGE ≥ 0.5 at Gauge #8 but exhibits unacceptable performance (KGE < -0.41) at nine gauges." Correspondingly, we have also revised similar discussions for flow depth (Lines 322-326) as "The downscaled flow depth achieves KGE ≥ 0.5 (good performance) at all gauges except Gauge #14, where small inundation occurs. In contrast, the LF simulation only achieves KGE ≥ 0.5 at two gauges but exhibits poor performance (KGE < -0.41; Knoben et al., 2019) at three gauges, whereas the performance at the other gauges is barely acceptable (-0.41 < KGE < 0.5). Notably, for four gauges (#3, #7, #8, and #19), the downscaling approach achieves KGE ≥ 0.9 (excellent performance)."

**L277: Same here and elsewhere applied.**

Response: Please see our above response.

**L277: "the LF simulation only shows better performance than the downscaled one at Gauge #8"**

Response: Revised as suggested.

**L283: This acronym (ERDR) has never been introduced.**

Response: Sorry for the mistake. This should be the error from dimensionality reduction $ER_{DR}$ which has been defined in the last paragraph of Section 2.1.

**L283: This acronym (ERLSG) has never been introduced.**

Response: Sorry for the mistake. This should be the error from the LSG model $ER_{LSG}$ which has been defined in the last paragraph of Section 2.1.

**L366: May be something is missing.**

Response: We have revised this sentence (Lines 493-494) as "The LF model used in this study is about 84 times faster than the HF model, which is significantly more efficient than the LF model adopted by Fraehr et al. (2023a) that is only 12 times faster".